# Suitability of a Low-Fidelity and Low-Cost Simulator for Teaching Basic Cardiopulmonary Resuscitation—“Hands-Only CPR”—To Nursing Students

**DOI:** 10.3390/nursrep15050162

**Published:** 2025-05-07

**Authors:** Zoila Esperanza Leiton-Espinoza, Ángel López-González, Maritza Evangelina Villanueva-Benites, Yrene E. Urbina-Rojas, Joseba Rabanales-Sotos, Yda Hoyos-Álvarez, María D. Pilar Gómez-Lujan

**Affiliations:** 1Faculty of Nursing, National University of Trujillo, Trujillo 13001, Peru; zeleiton@unitru.edu.pe (Z.E.L.-E.); mpgomezl@unitru.edu.pe (M.D.P.G.-L.); 2Nursing College, Universidad de Castilla La Mancha, 02006 Albacete, Spain; joseba.rabanales@uclm.es; 3Department of Clinical Sciences, Faculty of Nursing, National University of the Peruvian Amazon, Iquitos 16001, Peru; maritza.villanueva@unapiquitos.edu.pe; 4Faculty of Nursing, National University of Tumbes, Tumbes 24001, Peru; yurbina@untumbes.edu.pe; 5General Intensive Care Unit, Hospital Regional Docente of Trujillo, Trujillo 13001, Peru; mhoyos@unitru.edu.pe

**Keywords:** basic cardiac life support, heart massage, attitudes, practice, nursing degree students

## Abstract

**Objectives:** The objectives of this study were to determine the suitability of the low-fidelity/low-cost simulator “Salvando a Rosita^®^” in the acquisition of “hands-only CPR” skills by adult nursing degree students. **Methods**: A quasi-experimental quantitative study was carried out with a single experimental group that included 89 nursing degree students; it was conducted in November and December 2024 at the National University of Trujillo, Peru. **Results**: The STAI-TA score was 17.30, and the STAI-SA score was 37.00 points. Women showed a greater level of SA (*p* = 0.002). The required effort was described by women as being high and by men as light (*p* < 0.001). The compression rate was 125.7, and the percentage of compressions with an adequate depth was 89.6%. Overweight/obese individuals achieved more correct compressions (*p* < 0.01). The attitudes toward alerting emergency services, remaining calm while a person is in cardiac arrest, applying the CPR sequence automatically, and performing CCs were better after receiving training. The majority considered the “Salvando a Rosita^®^” simulator to be useful for teaching “hands-only CPR” to students in the first cycles of a nursing degree. **Conclusions**: The “Salvando a Rosita^®^” simulator was found to be an appropriate tool for teaching “hands-only CPR” to students in either the first cycles of health sciences or in other related professions.

## 1. Introduction

The survival rates associated with an out-of-hospital cardiac arrest (OHCA) continue to be low, ranging between 5 and 30% for various countries and regions [1]. The factors associated with survival after OHCA include the sociodemographic characteristics of the environment in which they occur [2], with a lower survival being reported when they occur in highly populated areas or areas with a large population dispersion [3]. They occur more often in less developed countries and in people with a low social status [4]. In addition, the lack of knowledge among the population concerning OHCAs and cardiopulmonary resuscitation (CPR) has been identified as a determinant of a worse survival prognosis [5,6,7,8,9].

To improve the population’s knowledge of how to respond to OHCAs, training programs have been proposed that focus on teaching citizens during initial, basic, and higher education, with promising results [10,11]. International organizations advocate for the simplification and universality of CPR, and to this end, they have implemented initiatives such as “hands-only CPR” [12], which recommends the rapid application of CPR by any person until the arrival of emergency medical services.

The efforts to spread awareness of CPR are ongoing. Novel methodologies and didactic materials have been proposed in the teaching of CPR to improve the training and retention of what has been learned. This includes several areas, for example, learning on the basis of the acquisition of competencies, the time spent, self-learning, the use of virtual reality mobile apps, etc. The challenge is to know how to train properly, in the shortest possible time and with the optimal use of resources [11,13].

All are useful ways to facilitate the widespread dissemination of CPR, but effective training requires a practical approach [14,15]. In this sense, low-cost, portable, resistant, and proven training simulators are essential for providing training on a large scale, in a natural environment, or even in places with limited economic resources available to future health professionals in their early training phases. However, there is still a lack of knowledge about the usability of and quality of teaching with these manikins in nursing schools [16,17,18,19].

“Salvando a Rosita^®^” (Figure 1) is an economical simulator for the practical training of “hands-only CPR”. This device is composed of a heart of plastic material, with a hardness similar to that of other adult CPR simulators, and it utilizes a feedback system that is activated when properly pressed [20]. The heart is placed on a figure drawn on a tapestry, on which the basic ERC-CPR algorithm is printed (Figure 2) [6].

At the top of the tapestry, the importance of CPR skills is indicated, a red box indicates the need to call the emergency telephone number, and in the centre, there is a green box with the Automatic external defibrillator (AED) pads and instructions on how to use it. On Rosita’s chest, the silhouette of the heart and the area where the AED patches should be placed are drawn. At the bottom of the tapestry the basic ERC-CPR algorithm is printed.

The aim of this research was to determine the suitability of the low-fidelity/low-cost simulator “Salvando a Rosita^®^” in the acquisition of “hands-only CPR” skills by adult nursing degree students at the National University of Trujillo.

## 2. Materials and Methods

A quasi-experimental quantitative study was carried out with a single experimental group (Figure 3) at the Faculty of Nursing of the National University of Trujillo (UNT), Peru. The participants performed “hands-only CPR” maneuvers on the “Salvando a Rosita^®^” simulator.

All the participants provided written informed consent. This study followed the ethical principles of the Helsinki Convention. The Research and Ethics Committee of the National University of Trujillo, Peru, approved the study protocol under code 01/2024/DIN-UNT.

### 2.1. Participants

The sample selection for this study was non-probabilistic and based on convenience, using a total of 148 accessible students in the nursing degree program at the UNT who were in their 2nd or 4th cycle. The inclusion criterion was regular enrollment in the 2nd or 4th cycles of the nursing degree program at the UNT. The exclusion criteria were suffering from an illness or disability that would contraindicate or prevent the performance of CPR maneuvers, not attending CPR training prior to the start of the investigation, and having prior knowledge of and/or training in CPR.

### 2.2. Procedures and Instruments

#### Intervention

Stage 1: An initial baseline assessment was performed on all the participants, consisting of the collection of personal and anthropometric data. Finally, the participants were scored on a scale that is used to measure the attitudes of nursing students toward “hands-only CPR” [21].

Stage 2: Subsequently, all the participants underwent a standardized theoretical session of “hands-only CPR”, followed by practice using the “Salvando a Rosita^®^” simulator using a four-step sequence whose objective is the development of psychomotor skills [22].

Stage 3: In the following days, all the participants performed 2 uninterrupted minutes of “hands-only CPR” on the “Salvando a Rosita^®^” simulator. At the end of the test, the participants were asked about their perception of the effort required, their level of anxiety, their opinion about the usefulness of the “Salvando a Rosita^®^” simulator, and their self-perception of the correctness of the CCs performed. Finally, they were scored on a scale that was used to measure the attitudes of nursing students toward “hands-only CPR” with the “Salvando a Rosita^®^” simulator [21].

In all the participants, the following variables were collected in addition to the sociodemographic age, sex, and degree cycle in which they were found:

Anthropometric variables:

Weight: Weight was measured as the average of two values assessed using an approved Tanita DC 430-SMA balance, with the individual being barefoot and in light clothing. The readings were rounded to 100 g.

Size: The size was measured as the average of two values assessed using a wall height rod (Seca-222), with the individual being barefoot, in an upright position, and making his or her mid-sagittal line coincide with the midline of the height rod. The head was placed parallel to the ground. The size was measured to the nearest millimeter.

Body mass index (BMI): The BMI was calculated as the quotient of the weight (kg) and height^2^ (m), with four categories defined as follows: low weight (<18.5 kg/m^2^), normal weight (18.5–24.99 kg/m^2^), overweight (25–29.99 kg/m^2^), and obese ≥30 kg/m^2^) [23].

Chest compressions:

To evaluate the quality of the CCs, the following formula was used: correct CC (%): [CC with adequate depth (%) + CC with adequate compression rate (%)/2] [24]. The depth of the CCs was considered correct when the following conditions were met:

The heart was pressed with sufficient force (28.5–69 kg), the auditory feedback system was activated, and the rate reached 100–120 compressions/minute [20].

Rating of perceived exertion (RPE): The Borg scale of perceived effort (Borg rating) was used. The Borg scale classifies the perceived exertion (light = 6–12; somewhat hard = 12–14; hard = 15–16; very hard = 17–20) [25].

Self-perception of the correctness of the CCs performed: This measure involved asking the participant if they believed they performed the CCs correctly. The possible answers were yes or no.

Opinion on the usefulness of the “Salvando a Rosita^®^” simulator: The participant was asked if he thought the simulator was useful for teaching “hands-only CPR”. The possible answers were yes or no.

State–trait anxiety inventory (STAI) questionnaire [26]: Questionnaire validated for the Spanish population, which studies anxiety in healthy adults. Self-administered questionnaire consisting of two scales, state anxiety (SA) and trait anxiety (TA), with 20 questions each, in a 4-point Likert response system according to intensity (0 = almost never/not at all; 1 = somewhat/sometimes; 2 = quite/often; 3 = a lot/almost always) that provides a numerical value ranging from 0 to 60 points. The total STAI is the sum of SA and TA.

### 2.3. Data Analysis

After the initial examination of the data, extreme values and outliers were identified, and the authenticity of the data was verified. The different variables were adjusted to a normal distribution.

To verify the fit to normal distribution of quantitative variables, we used statistics (Kolmogorov–Smirnov test). Descriptive characteristics of the participants were expressed as means ± standard deviation. Differences according to sex in the demographic and anthropometric variables were tested using Student’s *t*-test. The chi-squared test was used to compare sex differences between the BMI categories.

To evaluate whether the “Salvando a Rosita^®^” simulator was capable of discriminating the correctness of the CCs performed by the participants according to their BMI, the BMI was categorized according to the age and sex cutoff points defined by the World Health Organization. Differences in the %CC by the weight status categories (underweight -<18.5- vs. normal weight/overweight -≥18.5-) were tested using analysis of variance (ANOVA).

A significance level of *p* ≤ 0.05 was established for all analyses. All the statistical analyses were performed with the statistical package SPSS 26.0 (IBM SPSS Statistics, Armonk, NY, USA) [27].

## 3. Results

Table 1 shows the demographic and anthropometric characteristics of the 89 students included in the sample compared by sex. The usual anthropometric differences between men and women were observed, with higher BMI values for men. The normal weight category prevailed in women, and the overweight category prevailed in men.

The results of the STAI questionnaire by sex are shown in Table 2. The STAI-TA score was 17.30 ± 4.56 points, and the STAI-SA score was 37.00 ± 4.50 points. These values categorize the STAI-TA in men and women as below average and the STAI-SA as high. We found significant differences between the STAI-SA anxiety scores of men and women, with women showing a greater level of anxiety (*p* = 0.002).

Table 3 indicates that at the end of the two minutes of “hands-only CPR”, after performing approximately 240 compressions on the heart of the “Salvando a Rosita^®^” simulator, the effort was rated mostly as being high in terms of the perception of intensity and difficulty in carrying out the activity. The effort was described by women as high and by men as light, and these differences were statistically significant.

The rhythm and the percentage of CCs with adequate depth, as well as the characteristics of the remaining components of the CPR performed, are shown by sex in Table 4. The rate of compressions was 125.7 (120.7–130.8), and the percentage of CCs with adequate strength was 89.6% (83.9–95.3) without significant differences between men and women in both cases. In the bivariate analysis, statistically significant differences were observed according to the BMI categories, with those categorized as overweight/obesity reaching a higher % of suitable CCs (70.92). There were no differences in the various components of the CPR.

Table 5 shows that the attitudes toward alerting emergency services, remaining calm while a person is in cardiac arrest, applying the CPR sequence automatically, and performing CCs were greater after receiving training.

Finally, Table 6 indicates that the self-perception of having performed CCs correctly was 44.9. The majority of the men answered that they had performed CCs correctly (69.0) vs. 40.8% of the women, although this difference was not statistically significant. The participants mostly believed that the “Salvando a Rosita^®^” simulator is useful for teaching “hands-only CPR” to students in the first cycles of a nursing degree program.

## 4. Discussion

This study evaluated the suitability of the low-fidelity and low-cost “Salvando a Rosita^®^” simulator for teaching the “hands-only CPR” technique to students in the first cycles of a nursing degree program. To address the problem of the availability of commercial manikins, grass-roots movements have extended the idea of “do-it-yourself” and “low-cost manikins” [17,18]. “Salvando a Rosita^®^”, a low-cost simulator, demonstrated in the tests carried out in this study that it meets the quality standards required for teaching “hands-only CPR”. In general, the results showed that the perception of the effort made, the level of perceived anxiety, the percentage of correct CCs performed, and the attitude toward CPR after training are comparable to the results obtained using other simulators of a medium fidelity and a higher economic cost.

Stress arises in response to external pressure, whereas anxiety is the result of internal pressure. In this study, we set out to measure and compare the state anxiety (SA) and trait anxiety (TA) by performing “hands-only CPR” for two minutes with the “Salvando a Rosita^®^” simulator. The results showed that, in both men and women, the STAI-TA could be categorized as low, which indicates a low tendency to perceive situations as threatening. In contrast, the STAI-SA was high, reflecting a temporary emotional state of tension at the end of the maneuver. These results are similar to the findings of Maestre-Miquel et al. [28] and Martin-Conty et al. [29] in previous studies on CPR using medium- and high-fidelity simulators.

The mean RPE value reported by the participants after two minutes of “hands-only CPR” with “Salvando a Rosita^®^” was described as somewhat hard/hard, similar to that reported by other studies [26,30]. These results are comparable to those of Riera et al. [31], who reported that performing uninterrupted chest compressions for two minutes generated heart rates equivalent to 61% of the theoretical maximum achievable. This finding indicates that the level of perceived effort when using our simulator is similar to that experienced with other medium- and high-fidelity simulators.

The compression rate achieved with the “Salvando a Rosita^®^” simulator exceeded the range recommended by the 2021 guidelines (100–120 compressions per minute) by 5%. Since this simulator does not have a rate feedback system, the rescuers had to mentally set the rate after receiving prior instructions. The context of the test and the pressure caused by being evaluated may have contributed to this bias.

According to Taussig [32], a suitable compression rate in CPR ranges between 40 and 120 compressions per minute, ensuring a compression–decompression ratio of 50:50. Throughout the updates to the CPR guidelines, the recommended rate has progressively increased to the current range of 100–120 compressions per minute and could even continue to increase in the future [33].

Therefore, our findings suggest that, after training, the use of “Salvando a Rosita^®^” allows for compression rates close to those currently recommended [12] and within the values that could be proposed in future guidelines.

The percentage of CCs with the appropriate depth achieved in “Salvando a Rosita^®^” exceeded that reported in studies that used medium- or high-fidelity simulators [34,35]. This difference could be attributed to the measurement system used. In our case, the intensity of the feedback sound was measured with a sound-level meter, whereas in commercial simulators, the depth achieved is assessed by measuring the chest displacement. In addition, the ability to perform CCs with an adequate depth is related to the BMI of the rescuer; a BMI ≥18.5 in adults has been identified as a threshold of success, which largely coincides with our sample [36]. Consequently, “Salvando a Rosita^®^” discriminates against those who do not reach the adequate depth of compression according to their BMI. Values for the remaining CPR components have been reported by others [37].

Training in CPR is an integral part of the curriculum for health sciences students and other related professionals. Their learning guarantees patient safety through practical training in suitable simulation environments [38]. However, the materials used in the simulation may be economically unaffordable in various regions [39,40], and despite efforts to bring them closer to the most disadvantaged environments [41], they remain inaccessible in certain countries and areas [42].

The results of our research show that the quality indicators of “hands-only CPR” that are measured with other simulators, as well as other measured characteristics (the level of anxiety and the perception of effort), are achievable and measurable using “Salvando a Rosita^®^”. In this context, “Salvando a Rosita^®^” is perceived as a useful tool for teaching “hands-only CPR” to students in the first cycles of a nursing degree.

The educational curriculum in various countries provides CPR content at different educational stages, not only from a preventive perspective, but also from the point of view of learning techniques and protocols to be applied in critical situations. Nursing, as a teacher trainer field, will be key in this process of teaching CPR from an early age. The use of “Salvando a Rosita^®^” represents an alternative for training and learning CPR in schools and other community spaces, especially when commercial manikins are not available.

### 4.1. Implications for Practice

The nursing curriculum includes how to deal with in-hospital cardiac arrests. This teaching is performed with medium-/high-fidelity simulations. A first contact with “Salvando a Rosita^®^” will enable learning with individualized practice and basic techniques, such as assessing the situation, implementing initial safety measures, utilizing a technique to achieve correct CCs, etc. Public health strategies, including teaching the population how to act in situations of an OHCA, are worked into the same curriculum. Our simulator has proven to be an effective way for nursing professionals to teach CPR to target groups.

### 4.2. Strengths and Limitations

Among the limitations of “Salvando a Rosita^®^”, it is not suitable for learning situations where CPR quality is essential (final year undergraduate students and health science professionals). Another limitation could be the special predisposition that nursing students, from the first cycles of their training, show toward learning CPR in its different levels of complexity.

Further research is needed to demonstrate the suitability of “Salvando a Rosita^®^” for teaching “hands-only CPR” in other populations, such as schoolchildren, older adults, etc.

## 5. Conclusions

In the current sociosanitary context, where there is a high risk of OHCA victims lacking care in domestic, work, and social settings, the “Salvando a Rosita^®^” simulator demonstrates results that are comparable to those of other models. It is presented as an appropriate tool for teaching “hands-only CPR” to students in either the first cycles of health sciences or in other related professions, as well as citizens in general.

In light of the results obtained and having proven the suitability of the “Salvando a Rosita^®^” simulator as a useful tool in the teaching of “hands-only CPR” among initial-cycle nursing students, further research is suggested to evaluate the usefulness of the simulator for teaching CPR in other contexts and populations.

## Figures and Tables

**Figure 1 nursrep-15-00162-f001:**
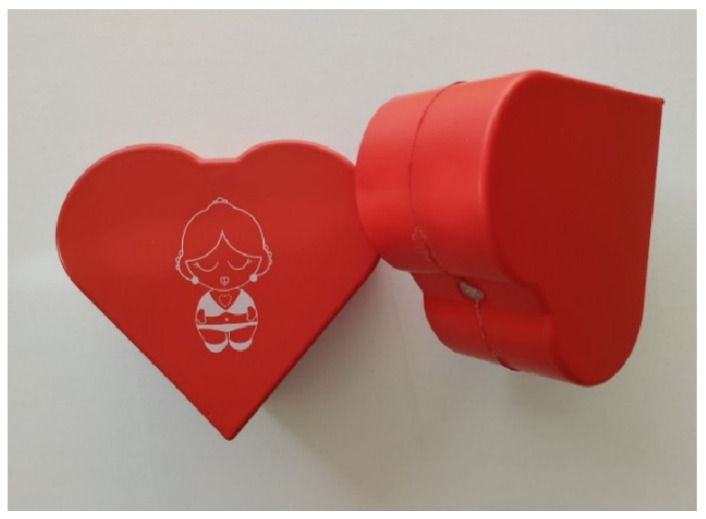
Heart of the auditory feedback simulator “Salvando a Rosita^®^”. Reproduced with permission from Leiton-Espinoza, Zoila E., Sánchez-Serrano, Susana and López-González, Ángel. Registro de obras literarias 00064-2019. 2019, INDECOPI, Peru.

**Figure 2 nursrep-15-00162-f002:**
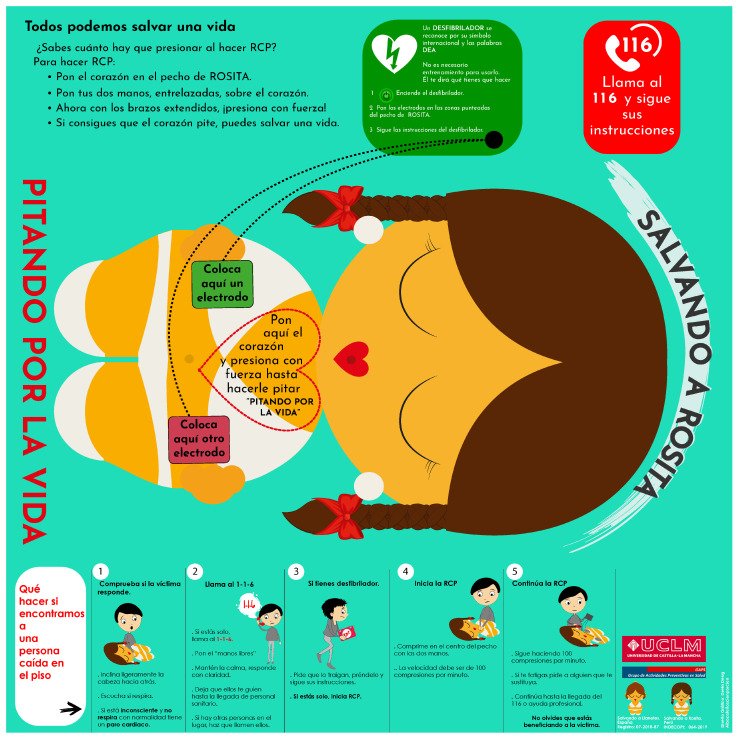
Auditory feedback simulator base for “Salvando a Rosita^®^”. Reproduced with permission from Leiton-Espinoza, Zoila E., Sánchez-Serrano, Susana and López-González, Ángel. Registro de obras literarias 00064-2019. 2019, INDECOPI, Peru.

**Figure 3 nursrep-15-00162-f003:**
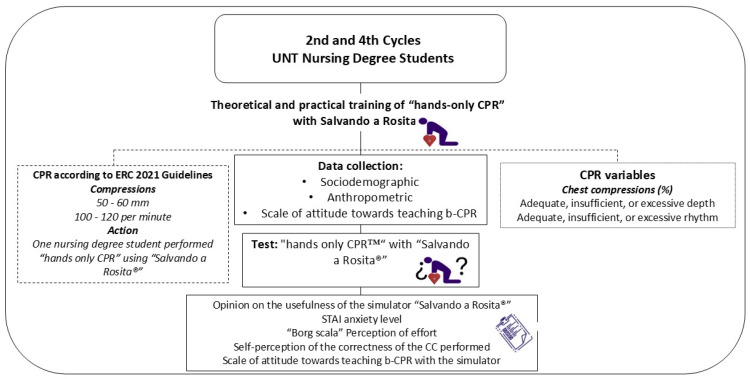
Study flow diagram.

**Table 1 nursrep-15-00162-t001:** Demographic and anthropometric variables of study population by sex.

	Total (n = 89)	Males (n = 13)	Females (n = 76)	*p*-Value
Age (years)	20.7 ± 1.88)	20.2 ± 1.88	20.8 ± 1.57	0.868
Weight (kg)	61.89 ± 11.25	72.45 ± 11.55	60.08 ± 10.23	0.386
Height (m)	157.3 ± 0.068	166.6 ± 0.055	155.7 (0.057)	**<0.001**
BMI (kg/m^2^)	24.94 ± 10.62	26.02 ± 3.40	24.76 (3.80)	0.267
Weight status **(%)**				
Normal weight	59.6	38.5	63.2	**<0.001**
Overweight/obesity	40.4	61.5	36.8	**<0.001**

Values are means ± standard deviation, except for weight status, which is shown as the prevalence. Bold type: *p* ≤ 0.05.

**Table 2 nursrep-15-00162-t002:** State–trait anxiety inventory for adults according to sex.

Variables	Total (n = 89)	Males (n = 13)	Females (n = 76)	*p*-Value for Males vs. Females
Trait Anxiety (TA)	17.30 ± 4.56	17.00 ± 5.38	17.00 ± 4.45	0.825
State Anxiety (SA)	37.00 ± 4.50	33.46 ± 4.53	37.60 ± 4.23	**0.002**

Values are means ± standard deviation. Bold type: *p* ≤ 0.05.

**Table 3 nursrep-15-00162-t003:** Borg rating by sex.

	Total (n = 89)	Males (n = 13)	Females (n = 76)	*p*-Value for Males vs. Females
Borg rating	14.43 ± 1.71	13.62 ± 1.71	14.57 ± 1.68	**<0.001**
Light	19.1	46.15	14.48	**<0.001**
Somewhat hard	44.9	23.08	48.68	**<0.001**
Hard	28.1	30.77	27.63	**<0.001**
Very hard	7.9	0	9.21	**<0.001**

Values are means ± standard deviation, except for categorization (in %). Bold type: *p* ≤ 0.05.

**Table 4 nursrep-15-00162-t004:** Differences in the correctness of the components of the CPR after 2 min by sex.

	Total (n = 89)	Males (n = 13)	Females (n = 76)	*p*-Value for Males vs. Females
Compression rate (CC/min)	125.7 ± 21.92	131.2 ± 18.95	124 ± 22.33	0.281
Correct compression depth (%)	89.6 ± 18.708	91.4 ± 11.55	87.9 ± 18.67	0.535
Normal weight	34.20 ± 29.04			
Overweight/obesity	70.92 ± 25.29			
*p*-value	<0.001			
Verify security of the environment	67.4 ± 0.471	53.8 ± 0.519	69.74 ± 0.462	0.338
Check consciousness	98.9 ± 0.106	100 ± 0.001	98.68 ± 0.115	1
Call emergency services	82 ± 0.386	92.31 ± 0.277	80.26 ± 0.401	0.449
Use correct hand position	85.4 ± 0.355	69.2 ± 0.376	64.47 ± 0.354	1
Use correct rescuer position	65.2 ± 0.479	84.62 ± 0.480	85.53 ± 0.482	1
Use adequate rate of CCs	38.2 ± 0.489	53.85 ± 0.519	35.53 ± 0.482	0.231

Values are expressed as average percentages ± standard deviation.

**Table 5 nursrep-15-00162-t005:** Attitude toward CPR before and after the training with “Salvando a Rosita^®^” by sex.

		CPR Pre-Training				CPR Post-Training				Total *p*-Value for Pre- vs. Post-Training
		Total (n = 89)	Males (n = 13)	Females (n = 76)	*p*-Value	Total (n = 89)	Males (n = 13)	Females (n = 76)	*p*-Value	
React and act to a person who does not respond	Very capable	40.4	38.5	40.8	0.320	58.4	76.9	55.3	0.324	0.071
	Somewhat capable	48.3	61.5	46.1		39.3	23.1	42.1		
	Not very capable	11.2	0	13.2		2.2	0	2.6		
Alert emergency services quickly	Very capable	69.7	76.9	68.4	0.743	77.5	84.6	76.3	0.697	**0.023**
	Somewhat capable	28.2	23.1	28.9		19.1	15.4	19.7		
	Not very capable	2.2	0	2.6		23.07	0	3.9		
Report details to the telephone operator of the emergency system	Very capable	57.3	53.8	57.9	0.645	69.7	92.3	65.8	0.155	0.545
	Somewhat capable	39.3	34.5	39.5		27	7.7	30.3		
	Not very capable	3.4	7.7	2.6		23.07	0	3.		
Assess whether a person is unconscious	Very capable	49.4	23.1	53.9	0.080	85.4	92.3	84.2	0.681	0.639
	Somewhat capable	48.3	76.9	43.4		11.2	7.7	11.8		
	Not very capable	2.2	0	2.63		23.07	0	3.9		
Recognize cardiac arrest	Very capable	22.5	15.4	23.7	0.777	47.2	53.8	46.1	0.274	0.296
	Somewhat capable	53.9	61.5	52.6		50.6	34.5	52.6		
	Not very capable	23.6	23.1	23.7		2.2	7.7	1.3		
Make decisions when faced with a person in cardiac arrest	Very capable	30.3	30.8	30.3	0.785	56.2	69.2	53.9	0.305	0.984
	Somewhat capable	47.2	53.8	46.1		43.8	30.8	46.1		
	Not very capable	22.5	15.4	23.7		0	0	0		
Stay calm when faced with a person in cardiac arrest	Very capable	48.3	61.5	40.8	0.288	64	92.3	59.2	0.70	**0.011**
	Somewhat capable	49.4	38.5	51.3		32.6	7.7	36.8		
	Not very capable	6.7	0	7.9		23.07	0	3.9		
Apply the CPR sequence automatically	Very capable	33.3	23.1	35.5	0.650	60.7	69.2	59.2	0.667	**0.045**
	Somewhat capable	37.1	46.2	35.5		36	30.8	36.8		
	Not very capable	29.2	30.8	28.9		23.07	0	3.9		
Indicate the need to open the airway in an unconscious person	Very capable	21.3	53.8	52.6	0.341	80.9	76.9	81.6	0.481	0.140
	Somewhat capable	39.3	38.5	40.8		14.6	23.1	13.2		
	Not very capable	39.3	7.7	6.6		4.5	0	5.3		
Indicate the need to assess whether an unconscious person is breathing	Very capable	52.8	53.85	52.63	0.981	88.8	100	86.8	0.382	0.086
	Somewhat capable	40.4	34.46	40.79		7.9	0	9.2		
	Not very capable	6.7	7.69	6.58		3.4	0	3.9		
Perform CCs	Very capable	39.3	38.5	39.5	0.716	52.8	69.2	50	0.374	**0.049**
	Somewhat capable	44.9	38.5	46.1		42.7	30.8	44.7		
	Not very capable	15.7	23.1	14.5		4.5	0	5.3		
Perform CPR on an adult	Very capable	36	30.8	36.8	0.856	44.9	69.2	40.8	0.140	0.229
	Somewhat capable	39.3	46.2	38.2		49.4	30.8	52.6		
	Not very capable	24.7	23.1	25		5.6	0	6.6		
Inform the emergency responders of what has been performed prior to their arrival	Very capable	58.4	46.2	60.5	0.181	83.1	92.3	81.6	0.622	0.174
	Somewhat capable	33.7	53.8	30.3		15.7	7.7	17.1		
	Not very capable	7.9	0	9.2		1.1	0	1.3		

Values are expressed as average percentages. Bold: *p* ≤ 0.05.

**Table 6 nursrep-15-00162-t006:** Perception of the percentage of correct compressions performed and the usefulness of the simulator by sex.

	Total (n = 89)	Males (n = 13)	Females (n = 76)	*p*-Value
Self-perception of having performed the compressions correctly	44.9	69.2	40.8	0.057
Belief that the simulator is useful for teaching BLS to first-year nursing students	97.8	100	97.4	0.554

Values are expressed as average percentages.

## Data Availability

The dataset is available from the authors upon request.

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
