# Peer review of "Suitability of a Low-Fidelity and Low-Cost Simulator for Teaching Basic Cardiopulmonary Resuscitation—“Hands-Only CPR”—To Nursing Students"

_nursrep, 2025, doi:10.3390/nursrep15050162_

Round 1
Reviewer 1 Report
Comments and Suggestions for Authors
Interesting study that addresses a fundamental content in CPR education, based on low-cost simulators.
This reviewer wants to applaud the authors for this approach. In order to improve this study for future publication, I offer some suggestions.
INTRODUCTION and METHODOLOGY
The introduction is correct and well-structured and concise, in addition to the well-described methodology. However, I recommend that when referring to the Borg scale, name that variable as rating of perceived exertion (RPE) according to Borg's scale, as it is the most commonly used term for perceived fatigue, regardless of the scale used.
RESULTS
In the results, I recommend designing the tables with the same format, according to the journal's style.
DISCUSSION
At the end of the discussion, I recommend including a small paragraph or section titled "implications for practice." This device needs to transcend beyond this study; this paragraph should address how nursing students could use it in their clinical or educational practice, for example, in the socio-health education of the population or in dissemination as a public health strategy.
Finally, strengths and limitations should not be a separate section; they should be a subsection of the discussion.
INTERESTING DOCUMENTARY SOURCES
I recommend reviewing these two articles, and if the authors deem it appropriate, include in the discussion the importance of LOW-COST devices for CPR education, whether they are DIY or manufactured at a very low price.
Do-It-Yourself Devices for Training CPR in Laypeople: A Scoping Review
Assessing the quality of chest compressions with a DIY low-cost manikin (LoCoMan) versus a standard manikin: a quasi-experimental study in primary education
Congratulations on the work.
Author Response
We thank the reviewer for his comments on the quality of the article and especially with regard to the contribution of the simulator. "Saving Rosita®" to the teaching of CPR. Your suggestions will contribute to a significant improvement of the manuscript.

Reviewer 2 Report
Comments and Suggestions for Authors The article presents a relevant and timely contribution to the field of health education, particularly in the practical training of nursing students in basic life support. The development and evaluation of the "Saving Rosita®" simulator stand out for their innovative, accessible, and easily applicable approach, especially in contexts with limited resources. I have provided some attached suggestions; however, the work is technically well-developed, methodologically well-conducted, and contributes to the expansion of the use of accessible simulators in CPR teaching. The suggested corrections are minor, aimed at improving clarity, coherence, and editorial consistency. In light of these considerations, I issue a favorable opinion for its publication, after the necessary adjustments.
Author Response

(The authors gave the same response as above.)

Reviewer 3 Report
Comments and Suggestions for Authors
Dear Authors
It has been a pleasure to have the opportunity to review your manuscript ”Usefulness of A Low Fidelity/Cost Simulator in the Teaching of Basic Cardiopulmonary Resuscitation “ Hands-Only CPR” in Nursing Students”
I would like to make some recommendations to improve the paper:
TITLE
Below, I'll give you some suggestions for changes, but I recommend you first address the rest of the sections and then return to this last point; it will be easier.
1. ”Usefulness of A Low Fidelity/Cost Simulator in the Teaching of Basic Cardiopulmonary Resuscitation “ Hands-Only CPR” in Nursing Students”. The title needs to be revised. The title addresses the "usefulness" of a simulator in training for performing hands-only CPR. However, after reading the results, I find that the aspects evaluated are: attitude beforehand, subjective perception of effort, self-perception of correct compressions, anxiety level after compressions, student opinion on the simulator's usefulness, and assessment of CPR quality. It's clear that the title doesn't reflect the content of the study. You focus on usefulness (the most subjective aspect expressed by the student), while neglecting the more objective and relevant aspects.
ABSTRACT
Below, I'll give you some suggestions for changes, but I recommend you first address the rest of the sections and then come back to this point; it'll be easier.
2. “Abstract: To determine the usefulness of the low fidelity/cost simulator "Salvando a Rosita®" in the acquisition of skills to apply for "Hands only CPR" in adults by students of the Nursing Degree. A quasi-experimental quantitative study was carried out with a single experimental group conducted from November and December 2024 Faculty of Nursing, National University of Trujillo, Perú”. The beginning of the summary must be modified, at this beginning you can add the objective or justification and the objective of the study, currently you have added the methodology that must be included in the METHODS section.
3. “A quasi-experimental quantita-tive study was carried out with a single experimental group conducted from November and December 2024 Faculty of Nursing, National University of Trujillo, Perú”. You must add: a) the study population (nursing students and the initial number under study); b) the sample, if it was already determined (89 students?); c) the ethics committee approval code.
4. “Results: The STAI-TA score was 17.30, and the STAI-SA score was 37.00 points. Women showing a greater level of anxiety SA (p = 0.002). Effort was described as something hard by women and by men as light p<0.001). The compression rate was 125.7, and the compression rate with adequate depth was 89.6%. Overweight/obese individuals achieved more correct compressions (p<0.01). Attitudes toward alerting emergency services, remaining calm before a person is in cardiac arrest, applying the CPR sequence automatically and performing CC were greater after receiving training. The majority considered that the ‘Saving Rosita®’ simulator is useful for students in the first cycles of Nursing Degree to learn ‘Hands-Only CPR’”. It is necessary to organize this section: 1) first, visualize the objective of the study (not the current one, with suggested modifications); 2) try to organize the results according to the objective or the study development process (methods).
5. “Conclusions: "Salvando a Rosita®" simulator, by demonstrating de-manding results comparable to those of other models, is presented as an appropriate tool for both students in the first cycles of Health Sciences and other related professions, as well as citizens in general, to learn “Hands only CPR””. Again, after reviewing the objective(s), try to address it in this section. Currently, your title is utility-oriented (see comments in the title section).
INTRODUCTION
6. It is recommended that the introduction be reviewed. As already mentioned, numerous student assessments are addressed. However, this introduction only refers to the importance of CPR. It is important to remember that it should focus on the educational aspect, which is where this research is developed, although it also provides a brief reference to the importance of this training in the social context. Nothing is mentioned regarding anxiety, attitude, effort, quality, etc. in students. It should place the reader of this future publication in a context of background on the topic studied later.
7. Figure 2, if available in English, please replace it so that the widest possible audience can understand it. If not, you can modify the text to English, specifying this in the description below the image. Improve the image quality.
8. Before the objective, I have not detected the specific JUSTIFICATION for the need to conduct this study. For example: As previously mentioned, it is necessary to have future health professionals with the knowledge and skills to perform basic CPR maneuvers. To do this, it is necessary to have trainers to facilitate these skills in simulation situations. For this reason, the following study is proposed. This will allow us to evaluate the quality of the CPR maneuver, but also to delve into the perceived effort, attitudes, and level of anxiety of the students...
9. “The aim of this research was to determine the usefulness of the low-fidelity/cost simulator "Salvando a Rosita®" in the acquisition of skills to apply "Hands only CPR" in adults by students of the Nursing Degree of the National University of Trujillo”. As previously mentioned, the results are not found in the title and objective lines. Therefore, it is recommended to review the objective. Perhaps you could create a general objective focused on the results of the training process and other specific ones. This part of the manuscript is of great importance; it must be aligned with the results and conclusions. From my experience, to measure the effectiveness of the intervention, it would be advisable to conduct a pre- and post-intervention assessment to measure the improvement in knowledge and skills. Therefore, you should identify the objective that best suits your study.
METHODS
10. The diagram indicates that students from the 2nd, 4th and 6th cycles participate, however in the section “2.1 Participants. A total of 148 students with a Nursing degree from UNT who were in the 2nd (n = 70) or 4th cycle (n = 78) were invited to participate.”. Unify criteria.
11. Based on the previous point, the study population would be 148 students from the 2nd and 4th cycles. Then, a representative population estimate is made that shows the need for a minimum of 86 students. Let's review the following statement: "The sample calculation was non-probabilistic in that to reach a confidence level of 0.5, a sampling error of 7% and, a confidence level of 95%, a final number of 86 students were included." There are significant errors. Let's break it down: 1) this type of calculation is performed for PROBABILISTIC sampling; 2) Regarding how to express it, it would be a confidence level of 95%, a sampling error of 7%, and an estimate of the population proportion of 0.5; 3) In addition, the result would be a random sample of 82 participants. program used for calculation? CHECK THE CALCULATIONS!
12. It is advisable to indicate in "Stage 3" the specific order in which the evaluations for each variable are performed to maintain the same order in the variables section and, at the same time, in the results section. Everything must be consistent chronologically. 13. 2.3 Study variables. “…Tanita DC 430-SMA balance…” . Add measuring range 14. 2.3 Study variables. “…a wall height rod Seca-222…” . Add measuring range 15. 2.3 Study variables. “BMI: calculated as the quotient of weight (kg) …”. Indicate the meaning of the acronym before the body mass index (BMI). Also, cite this index by Adolphe Quetelet. 16. 2.3 Study variables. “Subjective Perception of Effort (RPE). The Borg Scale Perceived Effort (Borg Rating) was used. Borg scale classifies perceived exertion (Light = 6-12; Somewhat hard = 12-14; Hard = 15-16; Very hard = 17-20)”. Remember to mention the Borg Scale. 17. 2.3 Study variables. “State-Trait Anxiety Inventory (STAI) questionnaire. The self-administered and vali-dated Spanish questionnaire was conceived as a research ….” Remember to mention the STAI questionnaire. 18. 2.4 Data Analysis. “The variables are described by measures of central tendency (mean) and dispersion (standard deviation)” place after the paragraph “After the initial examination ….”. 19. 2.4 Data Analysis. “As a criterion of bilateral statistical significance, p≤0.05 was used. All the statistical analyses were performed with IBM® SPSS® Statistics 26 software.”. Change for the statistical package SPSS 26.0 (IBM SPSS Statistics, USA) and cited.
RESULTS
20. As previously mentioned, follow an order in presenting the results, one that is related to "stage 3" of the methods section. The chronological process of measurements is important; therefore, if I understand this study correctly, attitude measurement was conducted before the training began, followed by several measurements after the training ended, along with the post-training attitude measurement.
DISCUSION
21. I recommend rereading the change in objective and the order in which the results were presented, and now, redoing the discussion in an orderly manner and comparing your results with the literature, preferably in an academic context.
CONCLUSIONS
22. Provide response to the new objective/new objectives.
I look forward to hearing from you and awaiting your replies and modifications.
Author Response

(The authors gave the same response as above.)

Round 2
Reviewer 3 Report
Comments and Suggestions for Authors
Dear Authors,
You have heeded all the suggestions provided to improve your manuscript.
I wish you the best of luck.
Best regards.